# Observation of Strong Interlayer Couplings in WS_2_/MoS_2_ Heterostructures via Low-Frequency Raman Spectroscopy

**DOI:** 10.3390/nano12091393

**Published:** 2022-04-19

**Authors:** Ki Hoon Shin, Min-Kyu Seo, Sangyeon Pak, A-Rang Jang, Jung Inn Sohn

**Affiliations:** 1Division of Physics and Semiconductor Science, Dongguk University, Seoul 04620, Korea; kihoonshin@dongguk.edu (K.H.S.); seominkyuu@gmail.com (M.-K.S.); 2School of Electronic and Electrical Engineering, Hongik University, Seoul 04066, Korea; spak@hongik.ac.kr; 3Department of Electrical Engineering, Semyung University, Jecheon 27136, Korea

**Keywords:** bilayer MoS_2_, bilayer WS_2_, WS_2_/MoS_2_ heterostructure, low-frequency Raman modes, interlayer coupling

## Abstract

Van der Waals (vdW) heterostructures based on two-dimensional (2D) transition metal dichalcogenides (TMDCs), particularly WS_2_/MoS_2_ heterostructures with type-II band alignments, are considered as ideal candidates for future functional optoelectronic applications owing to their efficient exciton dissociation and fast charge transfers. These physical properties of vdW heterostructures are mainly influenced by the interlayer coupling occurring at the interface. However, a comprehensive understanding of the interlayer coupling in vdW heterostructures is still lacking. Here, we present a detailed analysis of the low-frequency (LF) Raman modes, which are sensitive to interlayer coupling, in bilayers of MoS_2_, WS_2_, and WS_2_/MoS_2_ heterostructures directly grown using chemical vapor deposition to avoid undesirable interfacial contamination and stacking mismatch effects between the monolayers. We clearly observe two distinguishable LF Raman modes, the interlayer in-plane shear and out-of-plane layer-breathing modes, which are dependent on the twisting angles and interface quality between the monolayers, in all the 2D bilayered structures, including the vdW heterostructure. In contrast, LF modes are not observed in the MoS_2_ and WS_2_ monolayers. These results indicate that our directly grown 2D bilayered TMDCs with a favorable stacking configuration and high-quality interface can induce strong interlayer couplings, leading to LF Raman modes.

## 1. Introduction

Two-dimensional (2D) transition metal dichalcogenides (TMDCs), such as MoS_2_ and WS_2_, have attracted significant attention for flexible and transparent electronics and optoelectronics applications because of their superior mechanical, electrical, and optical properties, such as high mobility, good mechanical strength, superior transparency, and excellent flexibility, as well as strong light–matter interactions [1,2,3,4,5,6]. TMDCs have considered to be promising nanomaterials for nanoelectronics, optoelectronics, sensors, energy conversion, and energy storage devices over the past decades [7,8,9,10,11,12,13,14,15]. Recently, considerable efforts have been devoted to the vertical assembling and integration of distinct 2D TMDC monolayers into van der Waals (vdW) heterostructures with diverse band alignments to develop a wide range of optoelectronic devices [16,17,18]. Because the individual monolayers composing the heterostructure are held together only by weak vdW interactions, the physical properties of such devices are strongly dependent on the interlayer coupling that occurs at the interface of the TMDC-based vdW heterostructures [15,19,20]. Thus, understanding the underlying physics of the interlayer couplings in vdW heterostructures, which play an important role in determining the charge and energy transfer behavior, is crucial for designing and developing high-performance devices.

Among the various heterojunction configurations with the corresponding band structures, the WS_2_/MoS_2_ heterostructures are considered to be a promising platform for use in device applications to achieve a high device performance because of their type-II heterojunctions, which enable an efficient exciton dissociation and charge transport [21,22,23,24,25]. For instance, Tan et al. demonstrated that the interlayer coupling between WS_2_ and MoS_2_ can enable a long-lived trap state in WS_2_/MoS_2_ heterostructures, resulting in an enhanced photodetector performance with a large photoconductive gain and high responsivity [18]. Wang et al. reported that the strong interlayer coupling in WS_2_/MoS_2_ heterostructures reduces the energy intervals of electron transition, making it detectable under infrared light [19]. Thus, a fast and reliable method for characterizing the interlayer coupling within 2D TMDCs heterostructures is highly desirable for rapidly assessing the physics underlying the unique electronic structures and optical properties.

Raman spectroscopy is one of the most powerful nondestructive tools used for obtaining a detailed structural and electronic information on 2D layered TMDCs and vdW heterostructures [26]. In particular, it has been recently reported that low-frequency (LF) Raman modes can be used as an indicator to directly probe the interlayer coupling effects in vdW-based layered structures, including layered heterostructures, because the LF Raman vibrational modes originate from the weaker interlayer vdW restoring forces [27,28].

In a few studies, the interfaces of WS_2_/MoS_2_ heterostructures have been investigated using Raman scattering [29,30]. However, a detailed study on the interlayer coupling in WS_2_/MoS_2_ heterostructures through LF Raman analysis has not yet been conducted. 

In this study, we investigated and compared the LF Raman modes in the bilayers of MoS_2_, WS_2_, and WS_2_/MoS_2_ heterostructures using a confocal Raman spectrometer to clearly distinguish the LF Raman modes. Moreover, we directly synthesized the MoS_2_, WS_2_, and WS_2_/MoS_2_ heterostructures on a SiO_2_/Si substrate via chemical vapor deposition (CVD) to rule out the undesirable effects, such as mismatch in the stacking angle between the monolayers and interface contaminations, which generally result in physically stacked TMDCs [27,28,31]. The thicknesses of the directly grown mono- and bilayers were confirmed using atomic force microscopy (AFM). Raman spectroscopy measurements were performed to characterize the vibrational modes in the high-frequency and LF ranges. Two characteristic LF Raman modes, the interlayer in-plane shear (C) and out-of-plane layer-breathing (LB) modes, were clearly observed in all the 2D bilayered homo- and heterostructures, including the MoS_2_ and WS_2_ bilayers, and WS_2_/MoS_2_ heterostructures, whereas no LF modes were detected in the MoS_2_ and WS_2_ monolayers. These results indicate that the LF modes can be ascribed to the strong coupling in our samples without interface contaminations/defects and/or misalignment effects, compared to mechanically stacked samples. 

## 2. Materials and Methods

### 2.1. Sample Preparation 

First, a solution was prepared by adding MoO_3_ powder (200 mg, Sigma-Aldrich, St. Louis, MO, USA) and WO_3_ powder (200 mg, Sigma-Aldrich, St. Louis, MO, USA) to NH_4_OH (10 mL, 28–30% solution, Sigma-Aldrich, St. Louis, MO, USA) in a small vial. A 20 μL MoO_3_ and WO_3_ solution was dropped onto a substrate and spin coated at 3000 rpm for 1 min. The weight of the deposited MoO_3_ film and WO_3_ particles was found to be significantly small, around ~0.01 mg [32].

Bilayer MoS_2_ were grown using atmospheric pressure CVD (APCVD) (Figure 1a). The prepared MoO_3_ solution was coated on a 300-nm-thick SiO_2_/Si substrate and loaded into the center of a quartz tube furnace. A ceramic boat loaded with 100 mg of S powder was placed upstream of the furnace. The bilayer MoS_2_ growth was carried out at 730 °C for 10 min, and then the furnace was naturally cooled down to room temperature. 

Bilayer WS_2_ were grown using low-pressure CVD (LPCVD), in which the quartz tube was evacuated to a base pressure of approximately 10^−3^ Torr, and a mixture of Ar and H_2_ was flowed into the furnace. The prepared WO_3_ solution and S powder were placed in the same position as that fixed during the bilayer MoS_2_ growth process. The bilayer WS_2_ growth was performed at 850 °C for 10 min, and throughout the process, the furnace pressure was maintained at 15 Torr.

Next, a WS_2_/MoS_2_ heterostructure was grown using the LPCVD method. The substrates coated with the MoO_3_ and WO_3_ solution were loaded on the bottom and top of the crucible in a furnace, respectively. The S powder was placed in the same position as that fixed during the bilayer MoS_2_ growth process, and the growth temperature, time, and pressure were the same as those used in the bilayer WS_2_ growth process. As the temperature of the furnace increased, MoO_3_ was deposited and sulfurized on the substrate located at the top of the crucible. The temperature of furnace reached at 850 °C, and the vertical and lateral MoS_2_/WS_2_ heterostructures were synthesized [32]. 

### 2.2. Characterization 

The surface morphology and thicknesses of the CVD-grown bilayer TMDs were characterized using atomic force microscopy (AFM (Multimode 8, Bruker, Billerica, MA, USA)). 

Observing the LF Raman modes in 2D layered TMDCs and vdW heterostructures is challenging, because it is difficult to distinguish them from the Rayleigh line using a conventional Raman microscope [33]. In this study, Raman spectroscopy measurements were performed at room temperature using a confocal Raman spectrometer (alpha 300 M+, WiTech, Ulm, Germany). A green laser beam with a wavelength of 488 nm and power of approximately 1 mW was focused onto the individual samples by a 50× objective (NA: 0.55) with a long working distance, as illustrated in Figure 1d. The estimated laser spot size was 1.08 μm.

## 3. Results and Discussion

### 3.1. MoS_2_ Bilayers 

Monolayer and bilayer MoS_2_ were directly grown on a SiO_2_/Si substrate via the APCVD method, as shown in Figure 1a (see the Section 2 for more details).

Figure 1b,c show the AFM images of the triangular mono- and bilayer MoS_2_. The lateral size of both the layers was approximately 13–15 μm. The height profiles show that the thicknesses of the mono- and bilayer MoS_2_ were found to be approximately 0.8 and 1.7 nm, respectively.

Raman spectroscopy was performed to investigate the vibrational modes of the as-grown 2D bilayer MoS_2_, WS_2_, and WS_2_/MoS_2_ heterostructures. Figure 2a illustrates the four different vibrational modes with relative displacements between the metal and chalcogen atoms, which can be representatively observed in most few-layer TMDCs. These modes can be classified into two categories according to their frequencies: high-frequency and LF modes. In the high-frequency range (>300 cm^−1^), the E_2g_ and A_1g_ modes originate from the in-plane and out-of-plane atomic vibrations within the layers, respectively. It was observed that the E_2g_ mode softens (red-shift), while the A_1g_ mode stiffens (blue-shift) as the number of layers is increased. Consequently, the frequency difference between the two modes increases [34]. Because the high-frequency Raman modes of the mono- or multilayered TMDCs originate from the intralayer chemical bonds, the restoring forces are significantly affected by the strength of the intralayer bonding, resulting in less sensitivity to interlayer coupling [26,27,34,35]. In contrast, in the LF range (<50 cm^−1^), the interlayer in-plane shear (C) and out-of-plane layer-breathing (LB) modes are expected to have very low frequencies owing to the weak interlayer vdW restoring forces [36,37]. Thus, the analysis of the LF Raman modes can be treated as a more reliable method for directly probing the interlayer coupling in multilayered TMDCs.

Figure 2b shows the Raman spectra of the monolayer (black line) and bilayer MoS_2_ (red line), recorded in the high-frequency range of 320–480 cm^−1^. Note that the quality and lateral size of the MoS_2_ (and WS_2_) samples chosen for the Raman measurement were almost the same as those shown in Figure 1b,c. For the monolayer MoS_2_, two typical peaks were obtained at 386.7 and 404.7 cm^−1^, which correspond to the in-plane vibrational mode (E_2g_ mode) and out-of-plane vibrational mode (A_1g_ mode), respectively. Evidently, for the bilayer MoS_2_, the E_2g_ and A_1g_ modes shift in the opposite direction, and the frequency difference between these two modes increases from 18.1 cm^−1^ (for the monolayer MoS_2_) to 21.1 cm^−1^ [34,38,39]. 

Figure 2c shows the Raman spectra of the monolayer (black line) and bilayer MoS_2_ (red line), obtained in the LF range of −80 to 80 cm^−1^. Neither the C nor the LB modes are observed in the monolayer MoS_2_, as is generally expected for all the other monolayer TMDCs, because the interlayer restoring force is absent. In contrast to the monolayer MoS_2_, two distinct LF Raman peaks were observed in the spectra of the bilayer MoS_2_, as well as in the corresponding anti-Stokes spectra. A sharp peak at 40.2 cm^−1^ can be assigned to the LB mode of the bilayer MoS_2_. Another peak at 24.3 cm^−1^ can be assigned to the C mode of the bilayer MoS_2_, which is typically of lower frequency than that of the LB mode.

According to previous studies, the intensity and frequency of the Raman peaks corresponding to the C and LB modes in bilayer MoS_2_ depend on the twisting angles and interface quality between the top and bottom MoS_2_ layers. It has been reported that the C mode is clearly observed near 0° or 60°, and is even absent for certain twisting angles. [31] Thus, the clear observation of the C mode shown in Figure 2c implies that the bilayer MoS_2_ was grown with the 2H (60°) or 3R (0°) stacking configuration (as shown in Figure 1b,c), which is the most stable configuration of CVD-grown bilayer systems [40]. Furthermore, we observed that there exists only one peak corresponding to the LB mode. This finding indicates that unlike the mechanically stacked bilayer MoS_2_, the directly grown bilayer MoS_2_ had a uniform interlayer without localized strains, crystallographic defects, or wrinkles, which are usually introduced in mechanically transferred samples. Thus, these results suggest that our directly grown bilayer MoS_2_ can induce a strong interlayer coupling with highly ordered domains between the monolayers.

### 3.2. WS_2_ Bilayers 

Monolayer and bilayer WS_2_ were directly grown on a SiO_2_/Si substrate via an LPCVD method (see Section 2 for more details).

Figure 3a shows the Raman spectra of the monolayer (black line) and bilayer WS_2_ (red line), obtained in the high-frequency range of 320–480 cm^−1^. For the monolayer WS_2_, two typical Raman signals were observed at 356.5 and 417.5 cm^−1^, corresponding to the in-plane vibrational mode (E_2g_ mode) and out-of-plane vibrational mode (A_1g_ mode) of WS_2_, respectively_._ Moreover, the E_2g_ and A_1g_ modes of the bilayer WS_2_ shifted in the opposite direction compared to those of the monolayer WS_2_, resulting in an increase in the frequency difference between them, which is consistent with the previously reported results [41,42]. 

Figure 3b shows the Raman spectra of the monolayer (black line) and bilayer WS_2_ (red line), obtained in the LF range of −80 to 80 cm^−1^. Similar to the results of MoS_2_ shown in Figure 2c, for the monolayer WS_2_, no peaks corresponding to the C and LB modes were observed. However, two different peaks at 17.9, and 33.8 cm^−1^ were observed in the LF Raman spectrum of the bilayer WS_2_, which can be assigned to the C and LB modes in a Stokes Raman spectrum, respectively. These findings indicate that the LF Raman modes can be attributed to the interlayer coupling effect.

### 3.3. WS_2_/MoS_2_ Heterostructures 

A vertical WS_2_/MoS_2_ heterostructure, with WS_2_ on top of the MoS_2_ monolayer, was directly grown on a 300-nm-thick SiO_2_/Si substrate via the LPCVD method (see Section 2 for more details).

The Raman spectrum of the WS_2_/MoS_2_ heterostructure in the high-frequency range of 320–500 cm^−1^ is presented in Figure 4a. The characteristic peaks observed at 383.7 and 356.5 cm^−1^ can be ascribed to the E_2g_ modes of the individual MoS_2_ and WS_2_ monolayers, respectively, whereas those located at 404.7 and 418.2 cm^−1^ can be attributed to the A_1g_ modes of these individual MoS_2_ and WS_2_ monolayers, respectively, confirming the formation of the WS_2_/MoS_2_ heterostructure. 

In the LF range, as shown in Figure 4b, we also observed two sharp characteristic peaks at 19.5 and 33.8 cm^−1^, which can be assigned to the C and LB modes, respectively. Figure 4c shows the LF modes of the WS_2_/MoS_2_ heterostructure. In general, two physically transferred layers composed of vdW heterostructures are usually misaligned, leading to a lack of in-plane restoring force, thus resulting in the disappearance of the C mode. Furthermore, nonuniform interfaces with variable local stacking can result in the presence of multiple LB modes [27]. Unlike the observations made in a previous study [29], here, we observed that in the WS_2_/MoS_2_ heterostructure, only one peak existed corresponding to each C and LB mode, which were associated with a good stacking configuration and interface uniformity, respectively [31]. Notably, directly grown heterostructures with epitaxial interfaces can exhibit a strong interlayer coupling compared to the mechanically transferred heterostructures prepared using a dry-transfer method or exfoliated suspension drop casting [28,43]. Thus, we believe that the clear observation of the C and LB mode peaks can be attributed to the strong interlayer coupling arising from the epitaxially grown bilayers with a clear interface and stable configuration. 

## 4. Conclusions

In this study, we investigated the LF Raman modes to explore the interlayer coupling in the bilayered MoS_2_ structures, WS_2_ structures, and WS_2_/MoS_2_ heterostructures, which were directly synthesized via CVD. For all the 2D bilayered MoS_2_ and WS_2_ homostructures and WS_2_/MoS_2_ heterostructures, typical LF C and LB modes were observed, whereas no LB modes were detected in the MoS_2_ and WS_2_ monolayers. Moreover, we showed that the observed single LB mode and clear C mode peaks could be attributed to the high-quality homo- and heterojunctions with stable stacking configurations, which enable the induction of a strong interlayer coupling within the layers. Our results provide a fundamental understanding of the interlayer coupling in WS_2_/MoS_2_ heterostructures and other 2D TMDC-based vdW heterostructures and an ideal approach for designing and developing high-performance functional devices based on various vdW heterostructures. 

## Figures and Tables

**Figure 1 nanomaterials-12-01393-f001:**
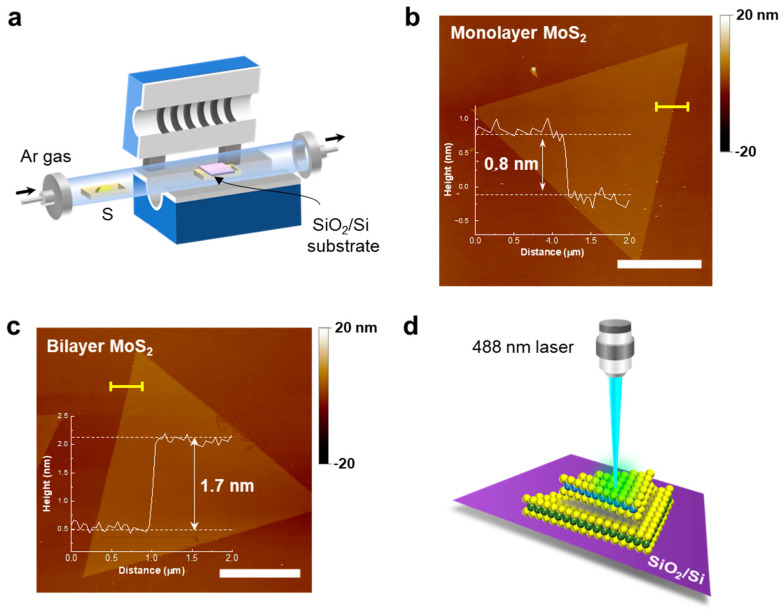
(**a**) Schematic of the growth process of the 2D bilayered MoS_2_, WS_2_, and WS_2_/MoS_2_ heterostructures via CVD. AFM topography images of the directly grown (**b**) monolayer and (**c**) bilayer MoS_2_. (Scale bar: 5 μm) (**d**) Schematic of the Raman spectroscopy measurements with a 488 nm laser focused on the vdW heterostructures.

**Figure 2 nanomaterials-12-01393-f002:**
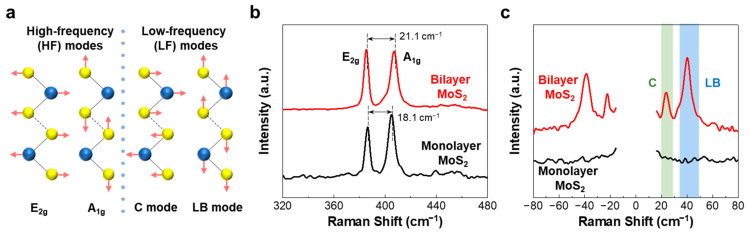
(**a**) Schematic of the lattice structure and four different vibrational modes of the bilayered TMDCs. Raman spectra of the monolayer (black lines) and bilayer MoS_2_ (red lines), obtained in the (**b**) high-frequency and (**c**) LF ranges. The Raman peaks enclosed by the green- and blue-colored regions correspond to the C and LB modes, respectively.

**Figure 3 nanomaterials-12-01393-f003:**
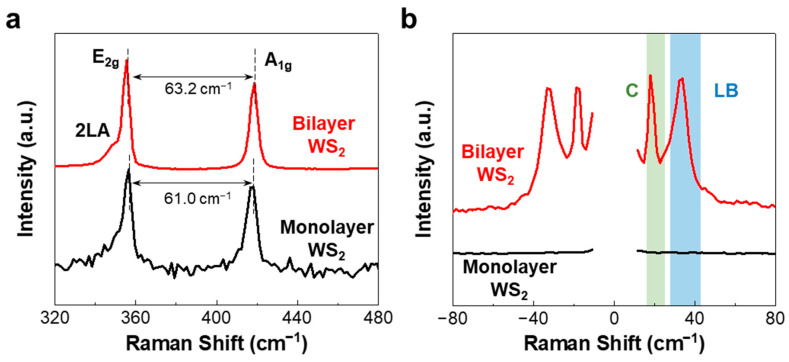
Raman spectra of the mono- (black lines) and bilayer WS_2_ (red lines), recorded in the (**a**) high-frequency and (**b**) LF ranges. The Raman peaks enclosed by the green- and blue-colored regions correspond to the C and LB modes, respectively.

**Figure 4 nanomaterials-12-01393-f004:**
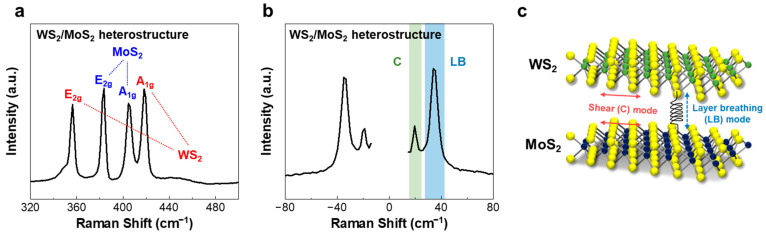
Raman spectra of the WS_2_/MoS_2_ heterostructures recorded in the (**a**) high-frequency and (**b**) LF ranges. The Raman peaks enclosed by the green- and blue-colored regions correspond to the C and LB modes, respectively. (**c**) Schematic depicting the interlayer interaction categorized by the C and LB modes in the LF range for the WS_2_/MoS_2_ heterostructure.

## Data Availability

Not applicable.

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
