# Peer review of "Observation of Strong Interlayer Couplings in WS2/MoS2 Heterostructures via Low-Frequency Raman Spectroscopy"

_nanomaterials, 2022, doi:10.3390/nano12091393_

Round 1

Reviewer 1 Report

The authors mainly investigated interlayer couplings in WS2/MoS2 heterostructures using low-frequency Raman spectroscopy. The motivation and explanation are sufficient, but some information about experiment procedure and results should be provided. The manuscript should be suitable and published in Nanomaterials after the authors revise the manuscript according to below comments.

1. In Section 2.1, Line 87-89, the authors should clearly present how the solution is prepared. What is the corresponding weight of MoO3/WO3 powder for MoS2/WO3 layer growth? In addition, for the WS2/MoS2 heterostructure growth in Line 100-102, how to coat the solution on the sample? Did the authors coat MoO3 and WO3 mixed solution and then sulfurize the sample at one time? Or the authors coated MoO3 solution and complete MoS2 sulfurization, afterward, they coated WO3 solution and sulfurized to complete WS2/MoS2 heterostructure?
The experiment procedure here should be described in detailed.

2. How large was the laser spot size when the authors carried out Raman measurement? That would be helpful for readers to understand the mechanism from the view of power density.

3. Line 110-121, the authors claimed that readers could see the Materials and Methods section to understand the growth procedures for monolayer and bilayer MoS2, however, the growth procedure for monolayer MoS2 was not presented. What is the main difference between monolayer and bilayer MoS2? Coating time, solution concentration, or sulfurization parameters?

4. The authors provided the AFM images only for monolayer and bilayer MoS2 to examine the layer number of MoS2. Is it possible to provide AFM images for monolayer, bilayer WS2, and the critical WS2/MoS2? That would help readers to know whole information about piece size, thickness, and stacking configurations. Especially the total thickness of WS2/MoS2 heterostrucrure would be a strong evidence to support the reasons such like good stacking configuration and interface uniformity for the appearance of C and LB modes observed in Raman results.

Author Response

April 12, 2022

Dear Alline Wang (Assistant Editor) and Reviewer #1

RE: Response letter for Manuscript No. nanomaterials-1668002

This response letter accompanies our online submission of the revised manuscript (Manuscript No. nanomaterials-1668002) “Observation of Strong Interlayer Couplings in WS2/MoS2 Heterostructures via Low-frequency Raman Spectroscopy.” By Shin et al. for publication in Nanomaterials.

 We would like to thank the referees for their constructive comments on our manuscript. The manuscript has been revised in response to reviewers’ comments. The following letter contains our detailed responses to the reviewers’ comments and all revisions made to the manuscript are marked red in the revised manuscript.

 Since we believe we made the necessary revisions according to all the referees’ comments, we now hope our revised manuscript is acceptable for publication in Nanomaterials.

Thank you.

Sincerely yours,

A-Rang Jang

Assistant Professor

Department of Electrical Engineering, Semyung University, Jecheon-si, Chungcheongbuk-do, Republic of Korea

Reviewer 2 Report

  1. More characterizations should (SEM, UV) be done
  2. What were the thicknesses of the MO3 and WO3 precursor films.
  3. How the separately coated MO3 and WO3 formed the interface. This is the most crucial question. It creates a doubt on the formation of heterostructures.
  4. Physical morphology of the heterostructure needs to be shown using cross-sectional HRTEM
  5. Raman peaks (figure 2 ) are not symmetric; you need to deconvolute and see what other materials can be there.
  6. In the introduction, more points should be added and cited related to Raman spectroscopy results obtained from various deposition methods.
  7. In fig 3(b), what are the various peaks' meanings, and what are they showing.
  8. English in the manuscript should be more polished.
  9. More information about the applications and history of MoS2 and WS2 should be added in the introduction part.
  10. AFM for the WS2 and heterostructure also should be shown.

Author Response

April 12, 2022

Dear Alline Wang (Assistant Editor) and Reviewer #2

RE: Response letter for Manuscript No. nanomaterials-1668002

This response letter accompanies our online submission of the revised manuscript (Manuscript No. nanomaterials-1668002) “Observation of Strong Interlayer Couplings in WS2/MoS2 Heterostructures via Low-frequency Raman Spectroscopy.” By Shin et al. for publication in Nanomaterials.

 We would like to thank the referees for their constructive comments on our manuscript. The manuscript has been revised in response to reviewers’ comments. The following letter contains our detailed responses to the reviewers’ comments and all revisions made to the manuscript are marked red in the revised manuscript.

 Since we believe we made the necessary revisions according to all the referees’ comments, we now hope our revised manuscript is acceptable for publication in Nanomaterials.

Thank you.

Sincerely yours,

A-Rang Jang

Assistant Professor

Department of Electrical Engineering, Semyung University, Jecheon-si, Chungcheongbuk-do, Republic of Korea

Round 2

Reviewer 2 Report

In response to almost all the queries author mentioning that it has been reported in the previous reports. If everything is reported earlier then this becomes a part submission. Please provide the cross-sectional TEM/SEM in the manuscript as it has not been reported in any one of your previous publications. 

Author Response

April 14, 2022

Dear Alline Wang (Assistant Editor) and Reviewer #2

RE: Response letter for Manuscript No. nanomaterials-1668002

This response letter accompanies our online submission of the revised manuscript (Manuscript No. nanomaterials-1668002) “Observation of Strong Interlayer Couplings in WS2/MoS2 Heterostructures via Low-frequency Raman Spectroscopy.” By Shin et al. for publication in Nanomaterials.

 We would like to thank the referees for their constructive comments on our manuscript. The manuscript has been revised in response to reviewers’ comments. The following letter contains our detailed responses to the reviewers’ comments and all revisions made to the manuscript are marked red in the revised manuscript.

 Since we believe we made the necessary revisions according to all the referees’ comments, we now hope our revised manuscript is acceptable for publication in Nanomaterials.

Thank you.

Sincerely yours,

A-Rang Jang

Assistant Professor

Department of Electrical Engineering, Semyung University, Jecheon-si, Chungcheongbuk-do, Republic of Korea
